# System for Monitoring Progress in a Mixing and Grinding Machine Using Sound Signal Processing

**DOI:** 10.3390/mi12091041

**Published:** 2021-08-29

**Authors:** Ekkawit Wangkanklang, Yoshikazu Koike

**Affiliations:** Electro-Mechanical System Laboratory, Department of Electronics Engineering, Shibaura Institute of Technology, Tokyo 135-8548, Japan; nb17102@shibaura-it.ac.jp

**Keywords:** Internet of Things, mixing and grinding machine, power spectral density, short-time Fourier transform, sound signal processing

## Abstract

In this paper we present a system for monitoring progress in a mixing and grinding machine via the signal processing of sound emitted by the machine. Our low-cost, low-maintenance system may improve automatic machines and the industrial Internet of Things. We used the Pumpkin Pi board and Raspberry Pi, which are low-cost hardware devices, for recording sounds via a microphone and analyzing the sound signals, respectively. Sound data obtained at regular intervals were compressed. The estimated power spectral density (PSD) values calculated from the sound signals were related to the status of the material during mixing and grinding. We examined the correlation between the PSD obtained by the STFT and the particle distributions in detail. We found that PSD values had both repeatability and a strong relation with the particle distributions that were created by the mixing and grinding machine, although the relation between the PSD and the particle size distributions was not merely linear. We used the PSD values to estimate the progress remotely during the operation of the machine.

## 1. Introduction

Mortars and pestles have traditionally been used to mix and grind materials, such as rice powder, peanuts, and pottery. However, machines are now used for mixing and grinding to produce large quantities of high-quality ground materials. In particular, the Ishikawa mixing and grinding machine is used in industrial manufacturing processes [1] for various types of materials, including those for electronic parts, chemical products, and art supplies, as well as some special materials, such as materials for solar batteries and fluorescent paint or gold powder.

Systems for monitoring mixing and grinding while the machine operates are critical. However, currently, operators cannot remotely observe the progress of mixing and grinding. Methods of checking the status of the material during the process, such as by using a microscope or conducting a sieving analysis, require the machine to be stopped to collect samples at regular intervals and are inefficient due to the increase in processing time and high cost of operation. Thus, the development of methods for remotely monitoring progress is highly desirable.

Advancements in the Internet of Things (IoT) have led to the development of online monitoring systems. For instance, Dhingra et al. proposed real-time monitoring using the Arduino integrated development environment software with a Wi-Fi communication module. The sensor network monitors air pollution or particles, and data are sent to a cloud system. Users can check pollution levels via an Android application, IoT-Mobair [2]. López-Vargas et al. proposed an online monitoring system on a website and a mobile application for solar measurement based on Arduino [3]. Lv et al. designed a monitoring system with an IoT technology for a smart city using a Zigbee wireless network for monitoring and controlling temperature, lighting, air conditioning, and various appliances [4]. Kumar et al. reported a lightweight, secure session key establishment scheme for smart home environments. They used a short authentication token and established a secure session key between a gateway network and smart devices [5]. To reduce home energy use, Han et al. monitored the energy consumption of home appliances with the Zigbee module and monitored the generation of renewable energy via power line communication [6]. For industrial manufacturing, Industry 4.0 is focusing on smart hardware and real-time data. Han et al. proposed a system for monitoring and predicting air pollution in the manufacturing industry. They designed sensors for measuring parameters such as carbon monoxide, nitrogen dioxide, sulfur dioxide, ozone, particulate matter, temperature, and humidity, and they collected the sensor data with the Zigbee network [7]. Zhang et al. investigated IoT technologies in manufacturing in order to obtain a system with an online monitoring architecture for steel casting [8]. Sung and Hsu reported an industrial real-time measurement and monitoring system using Zigbee and a data acquisition application. They designed and developed a linear variable differential transformer sensor, current sensor, carbon dioxide sensor, and energy monitor [9]. Deep learning can be utilized in future work, as claimed by [10]. This study proved that AI can be used to analyze and classify types of sound in a sample.

Various sensor technologies have been used to measure particle size. For example, Hu et al. used acoustic emission signals and signal analysis to measure the particle size of a pneumatic conveyer [11], and Carter and Yan [12] evaluated an electrostatic sensor and imaging sensor in order to measure the mass flow rate and size distribution of particles in a pneumatic suspension. Mao and Towhata monitored the particle size during the crushing process using acoustic emissions [13]. In addition, Carter et al. [14] measured particle size in a pneumatic suspension using digital image processing. However, these methods are complicated and inconvenient for monitoring progress in mixing and grinding machines.

Sound analysis has been widely used in many fields. For example, Takamichi et al. [15] created the CogKnife to identify types of food using a microphone sensor to record the cutting sound, which was analyzed by the system using spectrograms. Although their method could be used to evaluate the type of food with high accuracy, it could not be used to identify foods that produced little noise, such as tofu or jelly. Chitnaont et al. classified the sounds of a type of Thai flute called a khlui made from bamboo, pradu wood, and plastic [16]. The sounds were recorded by using a microphone and analyzed using spectral centroids. Bastari et al. proposed an acoustic signal processing system for measuring the particle size of coal powder. They studied the relationship between acoustic emission signals and powder particle size distribution [17]. Quino et al. monitored the sound of fibers breaking in E-glass fiber bundles by using Matlab and found that the system could detect the initiation and progression of failures [18]. De Cola et al. studied three types of sand grain shapes using sound measurements combined with microscopy [19]. Sen and Kumar Bhaumik reported a mathematical model of the sound emissions created during ball milling; however, the sound model is not suitable for mixing and grinding machines because it is limited to hard materials [20].

The mixing and grinding process has been employed for a long time in manufacturing processes. Recently, the demand for the application of IoT or deep learning to the mixing and grinding process has increased. For such extensions, a new and simple method of knowing the progress of manufacturing must be proposed, and it can be implemented in an existing mixing and grinding process without complicated remodeling or high costs. 

Here, we describe a method for estimating the progress in Ishikawa mixing and grinding machines using sound signal processing. The sound emissions are similar to noise; however, the noise level changes with the status of the grinding and mixing material. The sound is recorded and signal processing is performed. To validate the method, sieve analysis is used to determine the particle size distribution. The progress data are then compressed and sent through a wireless network to monitor the progress remotely. To extend the application to machine learning, we also prepared detailed data as training data. Therefore, we propose the combination of the STFT and PSD in the sound waveforms of the mixing and grinding process.

## 2. Materials and Methodology

### 2.1. Mixing and Grinding Machine

Figure 1 shows the mixing and grinding machine (model D101S, Ishikawa Kojo Co., Ltd., Tokyo, Japan), which simultaneously mixes and grinds and consists of a mortar, two pestles, a stirring bar, and an electrical motor. The machine rotates the pestles around the mortar with a gear rotation. The materials are mixed with the stirring bar and ground by the pestle tip as it spins around the bottom of the mortar. There are many types of ceramic mortars and pestles depending on the material being ground. For instance, a stone mortar and wood pestle are used for food processing, and a porcelain mortar and pestle are used for hard materials, such as those used in semiconductors. In this experiment, we used a porcelain mortar and pestle. The inverter in the machine for setting the motor speed could be set to 8–40 rpm by a controller, and the timer relay could be set to run the machine for periods of up to 12 h.

### 2.2. Pumpkin Pi

The Pumpkin Pi expansion board was developed by Marutsu, Japan, and is intended for experimenting with and measuring high-resolution audio signals. The Raspberry Pi model B+ does not have an analog input signal function and is disabled for voice recognition and measurement; therefore, the Pumpkin Pi board was developed to solve this problem. The Pumpkin Pi is an analog-to-digital (A-D) converter that consists of an 18-bit A-D converter (MCP3422, Microchip Technology Inc., Chandler, AZ, USA) for sensor measurements and an A-D converter (PCM1808, Texas Instruments, Dallas, TX, USA) for the analog input of 24 bit/s and 96 kHz used for audio measurements. The analog audio input has a maximum bit rate of 24 bit/s and a sampling frequency of 48 or 96 kHz via pulse code modulation (PCM). Figure 2 shows a diagram of the structure of the Pumpkin Pi.

### 2.3. Methodology

We used sound signal processing to monitor the progress of the mixing and grinding machine. A block diagram of the sound signal analysis is shown in Figure 3. The system consisted of recorded sound data, analysis signals, and average power spectral density (PSD) data. First, the sound level from the mixing and grinding process was recorded with an electric condenser microphone (ECMPC60, Sony Corp., Tokyo, Japan) with frequency responses between 50 and 15,000 Hz, low noise, and high sensitivity (−38 ± 3.5 dB). The microphone was installed in a cylinder to reduce powder adhesion and was placed on the back cover of the machine, 200 mm from the bottom mortar. The microphone recorded audio data and was connected via a 3.5 mm jack to the amplifier. Analog signals that passed through the amplifier (20–50 dB, AT-MA2, Audio-Technica Corp., Tokyo, Japan) were amplified with a gain of 35 dB. Second, we used the A-D convertor on the Pumpkin Pi board to convert the signal into a digital signal at a bit rate of 16 bit/s and a sampling frequency of 48 kHz. The Raspberry Pi board [21] was combined with the Pumpkin Pi board [22] to save data logs by recording the sound at regular intervals and analyzing the signal. We saved the data as WAV files and calculated the PSD using the short-time Fourier transform (STFT) [23] in Python on the Raspberry Pi. Finally, we compressed the data by calculating the average PSD value, which was shown in the real-time monitor, by using a transistor–transistor logic serial cable (FTDI-232R, Future Technology Devices International Ltd., Singapore). Owing to the compression of the data, we could remotely monitor the progress of the mixing and grinding process by using a wireless communication technology, such as WiFi. The PSD data that were received with a timestamp were employed a as database for the estimation of progress. Therefore, the conventional mixing and grinding machine, which did not have a sensing device for monitoring, could be connected to the IoT.

### 2.4. Estimating PSD

PSD estimation was used to analyze the sounds emitted from the mixing and grinding machine. The PSD is usually calculated with the Fourier transform by using parametric methods, such as an autoregressive model or moving average model, or by using non-parametric methods, such as a periodogram, Bartlett’s method, or Welch’s method [24]. We calculated the estimated PSD through a spectrogram by using the STFT to divide the analyzed signal into particular segments, such as frame lengths. Then, the Fourier transform was calculated and multiplied by a window function that overlapped at 50% to avoid spectral leakage. Many window function methods have been used, such as the Hanning window, Hamming window, Blackman, Gaussian, and Kaiser–Bessel methods. In general, the Hanning window has a good frequency resolution and reduced spectra. However, the Hamming window does a better job of cancelling the nearest side lobe [25]. It was confirmed that the PSD values obtained with the Hamming window were almost the same as the values obtained with the Hanning window. Therefore, we chose the Hamming window for the STFT and calculated the PSD values. The general equation for the STFT is:(1)$$x_{m}\left(f\right)=\sum_{n=-\infty }^{\infty }x\left(n\right)g\left(n-mR\right)e^{-j2\pi fn}$$
where *x_m_*(*f*) is the STFT, *x*(*n*) is a signal, *g*(*n*) is a window function, and *R* is the hop size. The hop size is the difference between the window length and the overlap length. The square of the magnitude of the STFT is a spectrogram, and we calculate the average PSD in decibels (dB) as:(2)$$average\;PSD\left(n\right)=mean\left(10\times \log_{10}(mean{\left|x_{m}\left(n,f\right)\right|}^{2}\right))$$

With Equation (2), we calculated the average frequencies and average times of the STFT. Finally, we calculated the average PSD values and collected the data in one file.

## 3. Experiment

Various types of materials can be processed by mixing and grinding. To test the sound signal processing, we used peanut butter, Japanese rice, and Japanese green tea. The conditions were as follows: the motor speed of the mixing and grinding machine was set at 20 rpm; the sound signal was recorded via a microphone for 5 s every 2 min; the wave file’s bit rate was 16 bit/s with a sampling frequency of 48 kHz; the average PSD value was calculated for the recorded data every 5 s.

### 3.1. Peanut Butter

We mixed and ground 80 g of roasted peanuts with 50 g of sugar. We observed the peanut butter from the beginning until it became creamy or until the end of the processing time. The machine ran continuously for 200 min. Figure 4a–d show the status of the material during mixing and grinding.

### 3.2. Japanese Green Tea

We mixed and ground 200 g of Japanese green tea. The mixing and grinding machine ran continuously for 180 min. Figure 5a–d show the status of the material during the mixing and grinding.

### 3.3. Rice Powder

We used 200 and 500 g of Japanese rice as a hard material. The mixing and grinding machine was operated continuously to grind the rice into powder. Figure 6a–d show the status of the 500 g of the material during the mixing and grinding operation.

## 4. Analysis and Results

### 4.1. Setup Position of Microphone

To set the position of the microphone, we measured the sound signal from the microphone with an oscilloscope and compared the signals from beside and behind the mixing and grinding machine with an empty crucible. The PSD values beside the machine were calculated as −5.0 dB/Hz, and those behind were found to be −4.2 dB/Hz. Because the signals were similar, we placed the microphone behind the cover of the machine (Figure 7b).

We also measured the machine’s noise at a rotation speed of 20 rpm. The spectrogram of the machine noise is shown in Figure 7c. The stationary components appeared around 12 kHz. Additionally, impulses appears around 1.5 and 2 s due to the motion of the pestle. However, the progress of the grinding and mixing affected such components as machine noise.

### 4.2. Waveform Analysis

Figure 8a,b show examples of the waveforms recorded for peanut butter at 30 and 60 min. The wave files contained 120,000 data points and had a sampling frequency of 48 kHz. The vertical axis represents sound pressure. The waveform graphs changed between 30 and 60 min. It was difficult to quantitatively analyze the differences in the signals between the recorded waveforms; therefore, the waveforms were converted into spectrograms (Figure 9).

### 4.3. Spectrogram Analysis

The relationship between the frequency and magnitude of the sound signals was analyzed. The color of the spectrogram shows the power level of the frequency: Yellow indicates a high value and blue indicates a low value. Figure 9a,b show spectrogram graphs of peanut butter at 30 and 60 min with a frequency between 0 and 15 kHz. The spectrogram is yellow between 0 and 2 kHz, which means that there was some noise while the microphone was recording. This is complicated to observe and monitor remotely. Therefore, the spectrograms were converted into PSD values with a frequency range of 2–15 kHz to estimate the PSD values.

### 4.4. PSD Estimation

We used the average PSD values to monitor the progress of the mixing and grinding machine. To make the PSD variations clear, we applied smoothing splines to the original curves. In the PSD variation graphs, both the original curves and those with the smoothing splines are shown. We examined the characteristics of the PSD values during grinding and mixing. We used peanuts, green tea, and Japanese rice and performed each experiment in duplicate (experiments 1 and 2) to verify the accuracy and reproducibility of the method.

Figure 10 shows the average PSD value for roasted peanuts and sugar. In experiment 1, the PSD value decreased to −18 dB/Hz, and the ground peanuts gathered into a ball. The PSD value increased to a maximum of −10 dB/Hz, and then decreased. After 40 min, the ball became bound together and became a large, sticky lump. Finally, the average PSD value decreased slowly until it approached a straight line as the lump of peanuts became creamy. In experiment 2, the PSD value decreased to −17 dB/Hz at 10 min, increased to a maximum of −9 dB/Hz at 40 min, and then decreased slowly until forming a straight line. The waveform graph for experiment 1 was similar to that for experiment 2 from the start until the maximum values, although subsequently, the PSD values were different. The difference arose because the temperature affected the viscosity and friction of the peanuts after the maximum PSD value was reached.

Figure 11 shows the average PSD values for Japanese green tea. In experiment 1, the estimated PSD value decreased from −16 to −20 dB/Hz within 60 min. The particle size of the green tea leaves decreased and the PSD value varied between −20 and −22 dB/Hz from 60 to 120 min, and some leaves became powder. After 120 min, the green tea was converted into powder. In experiment 2, the PSD value decreased to between −15 and −20 dB/Hz within 60 min, and then the value varied between −20 and −21 dB/Hz from 60 to 120 min, and some tea leaves became powder. After 120 min, the PSD value decreased to between −21 and −21.5 dB/Hz, which is the downward trend observed in the figure. The waveform graphs were similar for both experiments because the experiments were short and the PSD values decreased when the particle size decreased.

Figure 12 shows the average PSD values for 200 g of rice powder. To avoid small variations, we used the fifth-point-averaging filter curve to reduce the impulse signals. In experiment 1, the PSD value varied between −2 and −5 dB/Hz up to 8 h, and then increased to a maximum value of 1 dB/Hz at 8 h. The particle size of the rice decreased, and some of it became powder. The PSD values decreased to −8 dB/Hz at 12 h. In experiment 2, the PSD value varied between −3 and −6 dB/Hz. The PSD value increased to a maximum of −0.5 dB/Hz at 8 h, at which point the rice had mainly become powder. Finally, the PSD value decreased to −12 dB/Hz at 12 h, at which point all of the rice was powder. The waveform graphs were similar in both experiments from the start until the maximum value, and subsequently, the PSD values were different because the rice powder was affected by humidity.

Precision is quantified by repeatability and reproducibility; thus, we performed the Japanese rice powder experiment using 500 g of rice (Figure 13). The PSD values varied between −10 and −15 dB/Hz, and after 32 h, the rice particle size was smaller, and some rice had become powder. The PSD value increased to a maximum of 8.3 dB/Hz at 40 h, at which time the rice had mostly become powder. Finally, the PSD value decreased to −16 dB/Hz at 44 h, and all of the rice was transformed into powder. We found that the waveform graph for 500 g of rice powder (Figure 13) was similar to that for 200 g of rice powder (Figure 12).

### 4.5. Correlation between Particle Size and Sound Signal Processing

The particle sizes of the Japanese rice and green tea leaves were measured visually or with a microscope to determine the particle size distribution in order to acquire further information about the mixing and grinding process. The particle size distributions were obtained through a sieve analysis with sieve sizes of 250, 180, and 75 µm. The weights for each sieve size were recorded and the percentages were calculated. We calculated the correlation coefficient between the particle size distribution and the sound signal analysis by stopping the machine and collecting samples for calculating the size distribution at every 4 h for the Japanese rice and every 1 h for the green tea.

The equation for calculating the correlation coefficient is:(3)$$r=\frac{n(\sum xy)-(\sum x)(\sum y)}{\sqrt{[\sum x^{2}-(\sum x{)}^{2}][n\sum y^{2}-(\sum y{)}^{2}]}}$$
where *r* is the correlation coefficient, *n* is the number of pairs of data, *x* is the value of all sieve sizes, and *y* is the value of all average PSD values. We analyzed the correlation coefficient between the sieve sizes of 250, 180, and 75 µm and the average PSD values.

Table 1 shows the particle size distribution compared with the average PSD values of green tea leaves shown in Figure 14. For green tea leaves, the correlation coefficient between particle sizes of more than 250 µm and the average PSD value was 0.96; thus, the variables had a highly positive relationship. Particle sizes between 181 and 250 µm had a correlation coefficient of −0.96, and particle sizes smaller than 180 µm had a correlation coefficient of −0.88, meaning that the variables had a highly negative relationship.

Table 2 shows the particle size distribution compared with the average PSD values for 200 g of Japanese rice shown in Figure 12 (experiment 2). The correlation coefficient of rice with a particle size greater than 250 µm had an average PSD value of 0.65; thus, the variables had a positive relationship. Particle sizes between 181 and 250 µm had an average PSD value of −0.41, and particle sizes smaller than 180 µm had an average PSD value of −0.68.

Table 3 shows the particle size distribution compared with the average PSD values for 500 g of Japanese rice shown in Figure 13. The correlation coefficient of rice with a particle size greater than 250 µm had an average PSD value of 0.68; thus, the variables had a positive relationship. Particle sizes between 181 and 250 µm had an average PSD value of −0.61, and particle sizes smaller than 180 µm had an average PSD value of −0.58, indicating that the variables had a negative relationship. The experiments with 200 and 500 g of rice (Figure 12 and Figure 13) had similar correlation coefficients, indicating that the method was repeatable.

To employ the combination of the image of the curve of the PSD at a certain time, as shown in Figure 10, Figure 11, Figure 12 and Figure 13, machine learning could also be applied in our proposed method in order to make the accuracy of the estimation of progress the output. We plan to combine the images of the curve of the PSD with the measured particle distributions as training data in future.

## 5. Conclusions

We proposed a method for monitoring the progress in a mixing and grinding machine. To monitor data remotely, we used sound analysis and estimated the PSD. The PSD values for peanuts, green tea leaves, and Japanese rice were different. The PSD values for peanuts decreased at the beginning of the process and increased when the peanuts and sugar formed a lump, and then decreased again when the mixture became creamy. The process of mixing and grinding Japanese rice took a long time. The PSD values of Japanese rice did not change much until the rice grains began to crack and became powder; then, the PSD values began to increase. The PSD values increased to the maximum when the rice became mainly powder, and then the values decreased until all of the rice became powder. The PSD values of green tea leaves decreased as the particle size decreased. Regardless of the material and the mixing and grinding process, the PSD values decreased until the steady state, at which point the PSD did not change. Therefore, the PSD values were estimated as the materials were ground, and temperature and humidity were found to be important.

We measured the particle size distribution for Japanese rice and green tea and calculated the correlation between the particle distribution and the processed sound signal. The average PSD values depended directly on the particle size for Japanese green tea. However, for Japanese rice, the average PSD values increased to the maximum, and then decreased as the particle size decreased until it became a powder. The sound signal was able to indicate the progress in the mixing and grinding machine, and we demonstrated that the results were repeatable.

Our paper contributes greatly to the extension of the Ishikawa mixing and grinding machine with respect to automatically knowing the progress in manufacturing made by such a mixing and grinding machine. Additionally, the parameters that we obtained have possibilities for applications not only in the IoT, but also in machine learning. In the future, we plan to measure more of the same data indicated in this paper and to apply machine learning to the estimation of progress.

## Figures and Tables

**Figure 1 micromachines-12-01041-f001:**
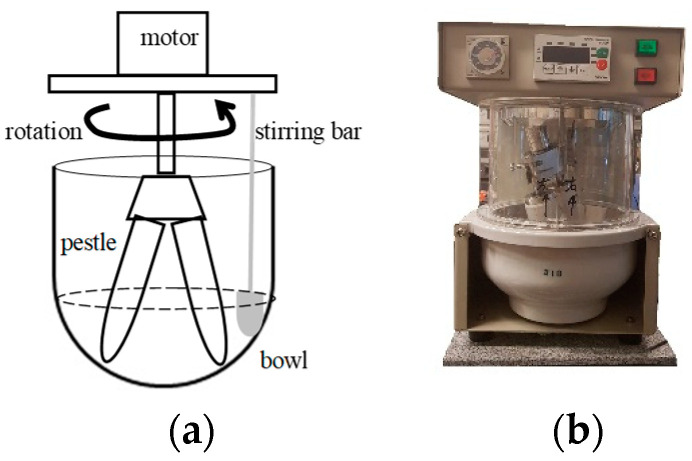
Mixing and grinding machine. (**a**) Schematic of the machine, (**b**) Whole picture of the machine.

**Figure 2 micromachines-12-01041-f002:**
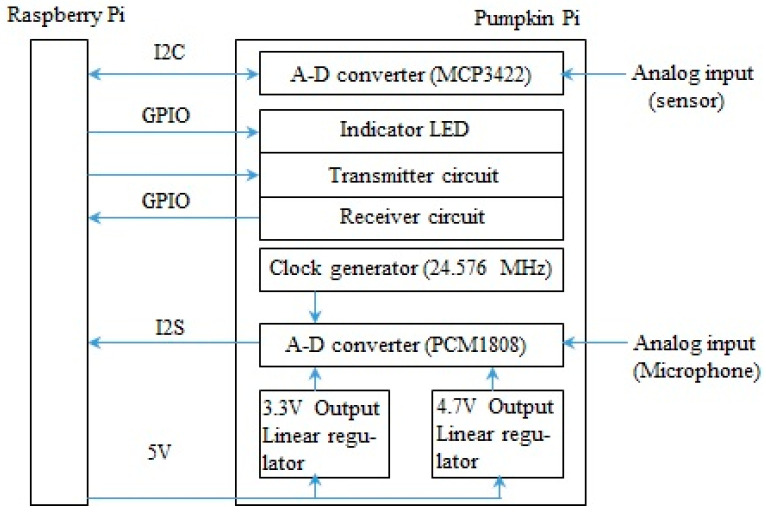
The structure of the developed Pumpkin Pi.

**Figure 3 micromachines-12-01041-f003:**
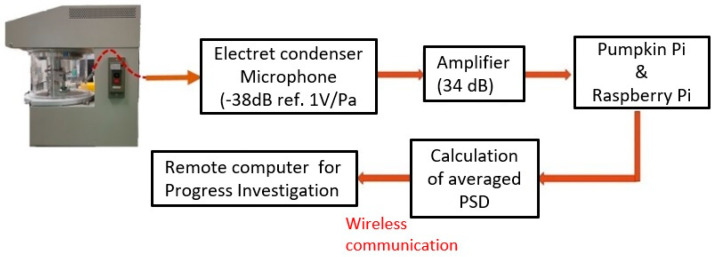
Block diagram for sound signal analysis.

**Figure 4 micromachines-12-01041-f004:**
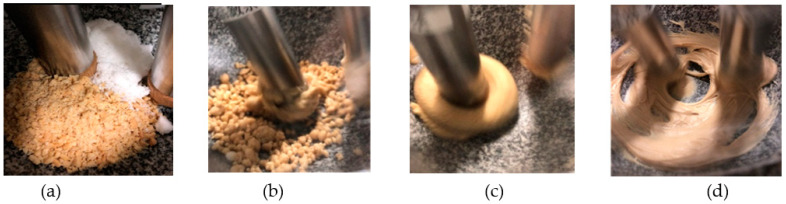
The progress of peanut butter during the experiment: (**a**) 0, (**b**) 20, (**c**) 40, and (**d**) 180 min.

**Figure 5 micromachines-12-01041-f005:**
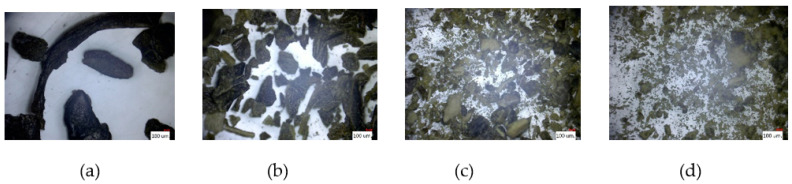
Progress of Japanese green tea during the experiment: (**a**) 0, (**b**) 60, (**c**) 120, and (**d**) 180 min.

**Figure 6 micromachines-12-01041-f006:**
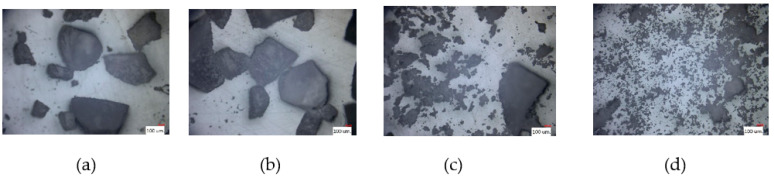
Progress of rice powder during the experiment: (**a**) 4, (**b**) 16, (**c**) 32, and (**d**) 48 h.

**Figure 7 micromachines-12-01041-f007:**
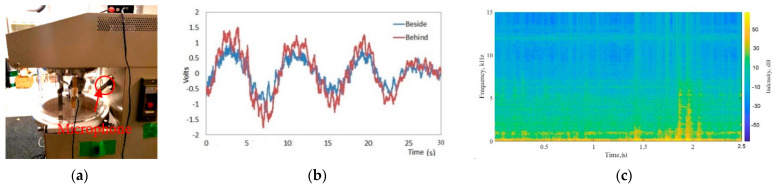
Microphone position (**a**), setup the microphone, (**b**) sound signals between behind and beside of the machine, and (**c**) spectrogram of motor with the pestle in the machine.

**Figure 8 micromachines-12-01041-f008:**
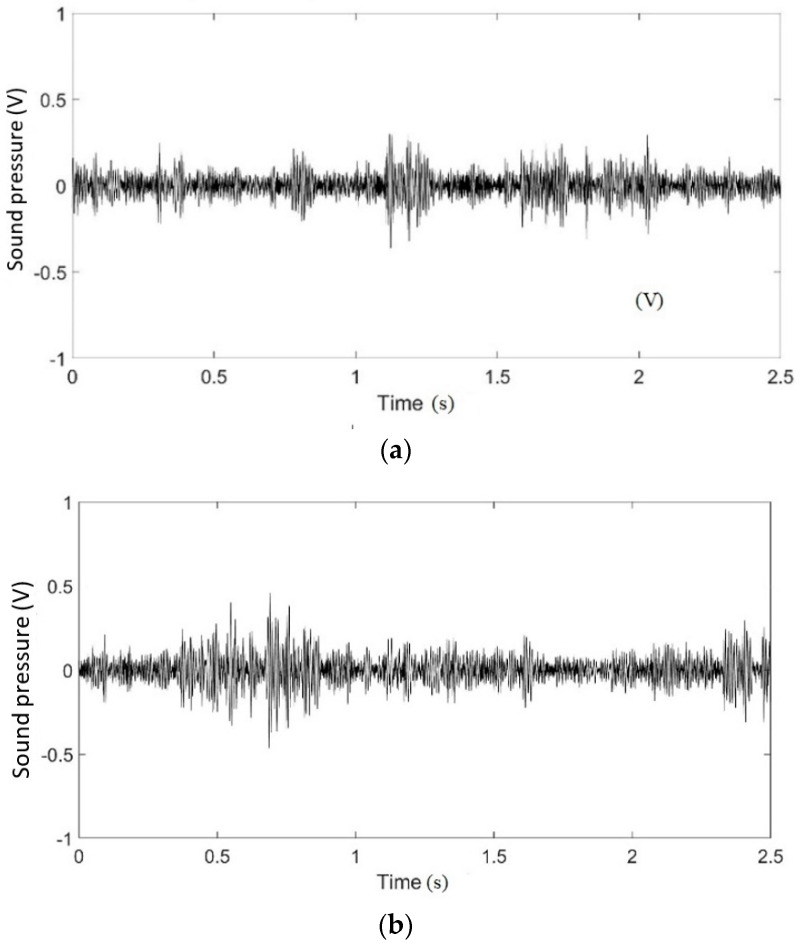
Recorded waveforms of peanut butter at (**a**) 30 and (**b**) 60 min.

**Figure 9 micromachines-12-01041-f009:**
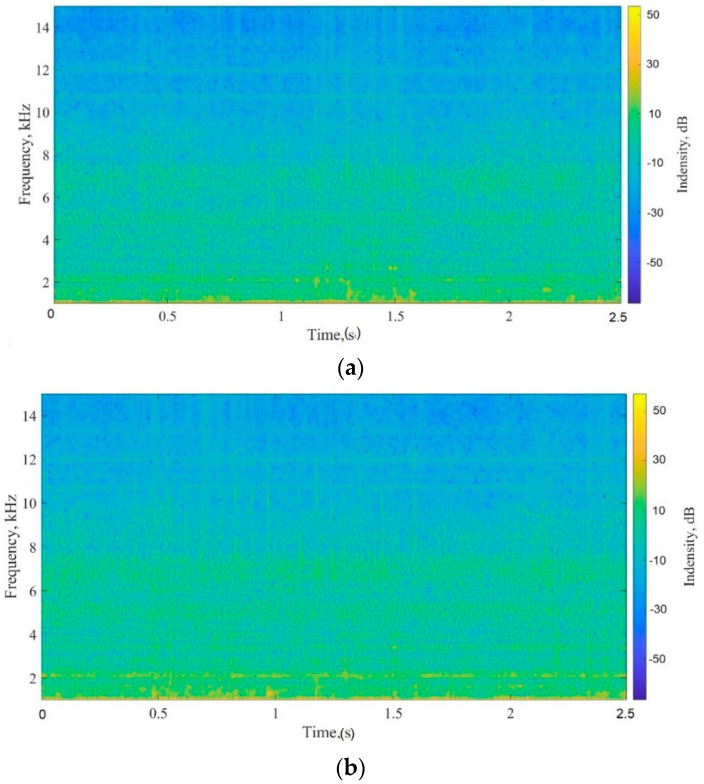
Converted spectrogram graphs of peanut butter at (**a**) 30 and (**b**) 60 min.

**Figure 10 micromachines-12-01041-f010:**
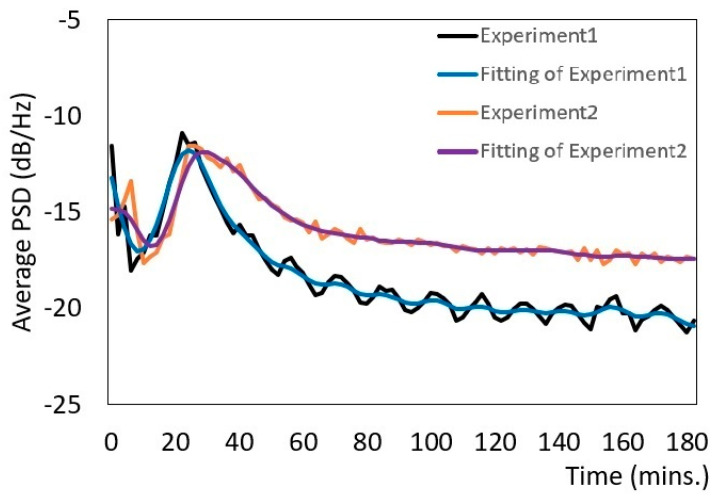
Variation of the PSD for peanuts.

**Figure 11 micromachines-12-01041-f011:**
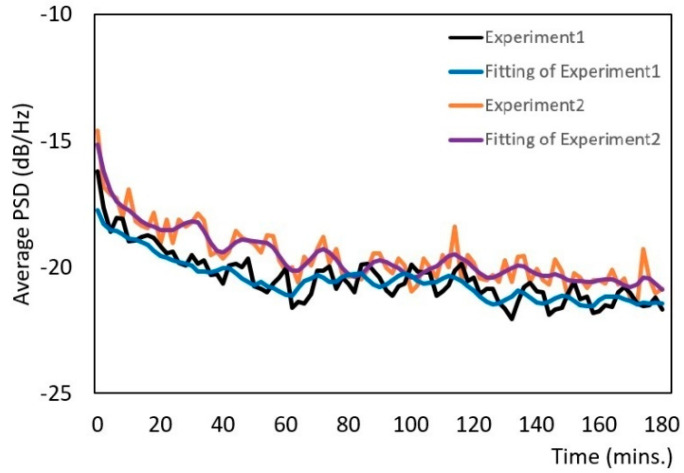
Variation of the PSD for Japanese green tea.

**Figure 12 micromachines-12-01041-f012:**
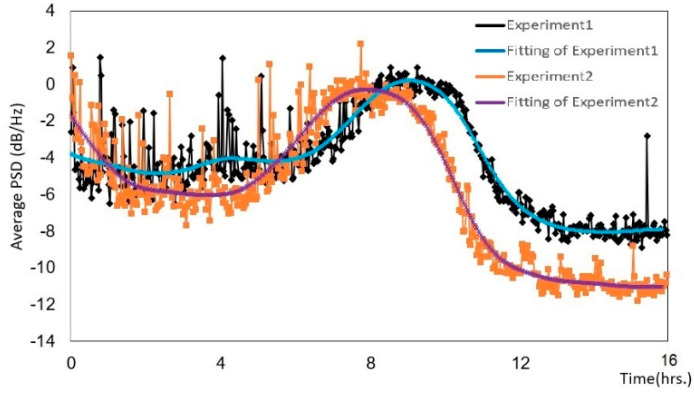
Variation of the PSD for 200 g of rice powder.

**Figure 13 micromachines-12-01041-f013:**
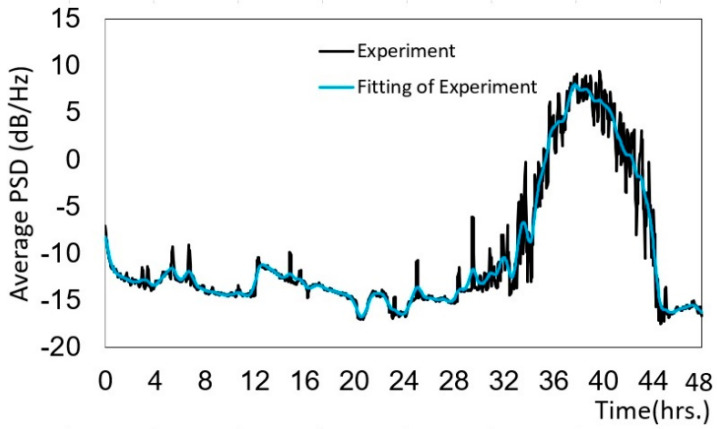
Variation of the PSD for 500 g of rice powder.

**Figure 14 micromachines-12-01041-f014:**
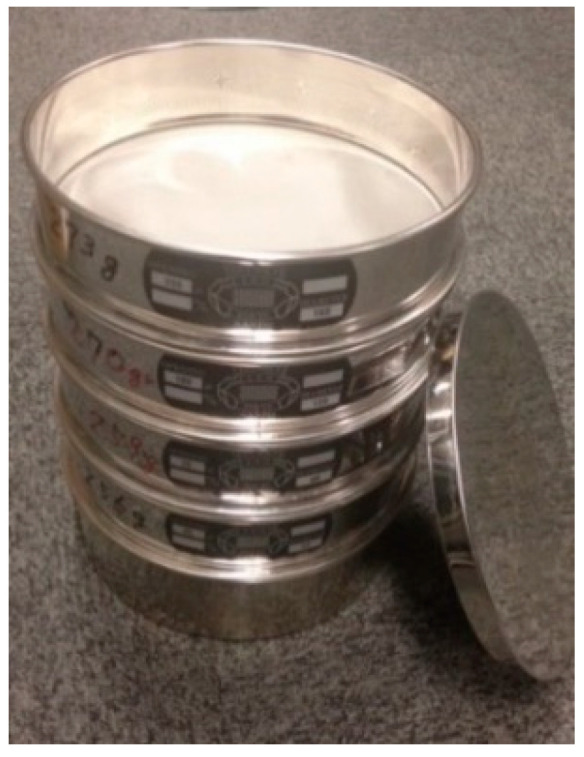
Sieving equipment.

**Table 1 micromachines-12-01041-t001:** Particle size percentage and average PSD values for Japanese green tea.

Time (h)	Sieve Size 250 µm (%)	Sieve Size 180 µm (%)	Sieve Size 75 µm (%)	Average PSD Value (dB/Hz)
0	99	0.3	0.25	−14.58
1	69.5	26.6	3.5	−19.72
2	50.9	41	4.4	−20.44
3	48.9	45	5.4	−21.13
4	41	50.5	8.2	−21.21

**Table 2 micromachines-12-01041-t002:** Particle size percentage and average PSD values for Japanese rice (200 g).

Time (h)	Sieve Size 250 µm (%)	Sieve Size 180 µm (%)	Sieve Size 75 µm (%)	Average PSD Value (dB/Hz)
0	100	0	0	−1.8
1	97.5	2.5	0.0	−3.6
2	96.6	3.4	0.1	−5.7
3	95.6	4.3	0.1	−5.8
4	94.7	5.0	0.3	−5.8
5	93.9	5.9	0.2	−5.3
6	91.8	7.8	0.4	−3.6
7	83.3	16.4	0.4	−1.0
8	61.5	37.2	1.3	−0.5
9	49.9	50.0	0.1	−0.9
10	28.9	56.2	14.9	−3.4
11	25.3	49.6	25.1	−8.3
12	21.7	61.1	17.2	−9.8
13	20.0	67.6	12.4	−10.7
14	17.3	66.0	16.8	−10.7
15	14.3	52.6	33.1	−11.1
16	10.9	59.9	29.2	−11.0

**Table 3 micromachines-12-01041-t003:** Particle size percentages and average PSD values for Japanese rice (500 g).

Time (h)	Sieve Size 250 µm (%)	Sieve Size 180 µm (%)	Sieve Size 75 µm (%)	Average PSD Value (dB/Hz)
0	100	0	0	−9.1
4	97	2	0	−13.1
8	96	2	0	−13.8
12	96	2	1	−13.4
16	94	4	2	−13.1
20	92	5	2	−15.1
24	90	4	3	−16.3
28	88	6	4	−14.8
32	86	8	4	−10.8
36	62	25	11	2.6
40	37	38	21	6.7
44	28	45	26	−12.7
48	21	51	26	−15.9

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
