# Peer review of "System for Monitoring Progress in a Mixing and Grinding Machine Using Sound Signal Processing"

_micromachines, 2021, doi:10.3390/mi12091041_

Round 1

Reviewer 1 Report

This paper mainly uses vibration sound detection to realize the processing status. This technology has basically been known for many years, and recently there are many detection technologies for sound detection, especially for automated manufacturing machines, so this paper is not sufficiently state-of-the-art. It is recommended to add some novel practices to improve the paper.

Author Response

Dear Reviewer,

      The attached file is my response paper. Thank you for your time and I am looking forward to your response.

Kind regards,
W.Ekkawit

Reviewer 2 Report

The manuscript presents an algorithm for monitoring progress in a mixing and grinding machine based on some well-known techniques such as STFT and PSD. However the combination of these methods seems to be unique for this particular application. Therefore the authors should be more clear and better stress the novelty of their work. 

The authors should also better explain why they use STFT and PSD of the sound signal and why that turns out to be more accurate than some others (e.g. wavelet transform) if tested at all.

Please explain why the Hamming window was used and wha it is more appropriate for this application. What were the result with other windows if tested at all?

In overall the contribution of the manuscript is not so trivial at all but it still needs a revision.

Author Response

(The authors gave the same response as above.)

Round 2

Reviewer 1 Report

1. The author has repeatedly emphasized the contribution of this paper to mechanical learning, but the content obviously does not discuss the machine learning method. It is not clear what is the relationship between input and predicted output in your idea.
2. The results show that the PSD value is related to the grain roughness of the rice grains, and it may also be possible to evaluate the feasibility by data fitting. It is recommended to add the content of the data fitting results, which can show the important findings of this article.
2. If xm(f) is a discrete STFT, it should be represented by xm(n,f). Similarly, Eq. (2) also needs to be corrected, EX. average PSD: Psd(n), or do you mean to average xm(n,f) in two dimensions?
3. As Figure 8, it seems that the vibrating audio signal is not steady-state continuous. Therefore, assuming that the average PSD is calculated in a limited time, it is likely that there will be a numerical deviation without a collision.
4. The CC: x and y of Eq. (3) do not specify the definition. In addition, the content that follows does not mention the CC value.
5. Figure 9 suggests adding a display threshold setting to remove noise, please refer to (https://www.mathworks.com/help/signal/ref/spectrogram.html). So that the result can be clearly expressed. In addition, the yellow sensation mentioned on page 7 could be the resonance of the rotating shaft of the container, or the noise of the motor rotation speed, which can be explained in detail. If the collision is not detected, when the speed changes, it is likely to affect the results.
6. The detailed way to doing IoT and the database of measurement data should be expressed. Pumpkin Pi is just a tool.

Author Response

Dear Reviewer,

        The attached file is my response paper.

Best regards,

W.Ekkawit

Round 3

Reviewer 1 Report

In my opinion, the revised manuscript does not have a strong or enough novelty for me. I would suggest this paper can't be published in Micromachines.

Moreover, I think the author doesn't answer my comments and revise their manuscript very carefully. Please see the attached file, I mark a lot of ignored comments in yellow color. 

Author Response

Dear Reviewer,

       The attached file is a response reviewer.

Best regards,
W.Ekkawit.
